# Suillin: A mixed-type acetylcholinesterase inhibitor from *Suillus luteus* which is used by Saraguros indigenous, southern Ecuador

**José Miguel Andrade, Pamela Pachar, Luisa Trujillo, Luis Cartuche** *

Departamento de Química, Universidad Técnica Particular de Loja (UTPL), Loja, Ecuador

* lecartuche@utpl.edu.ec

**Data Availability Statement:** All relevant data are within the paper and its Supporting Information files.

## Abstract

*Suillus luteus* (L.) Roussel is an edible mushroom commonly known as slippery jack or "Kallampa" by indigenous people from Loja province. It is used in traditional medicine to manage gastrointestinal disorders and headaches. In addition, edible mushrooms have been used for neurodegenerative diseases; however, there is no report about the anticholinesterase effect produced by this species. The aim of this work was to isolate the main secondary metabolite of *Suillus luteus* and characterize its inhibitory potential against acetylcholinesterase. Fruiting bodies were extracted with ethanol (EtOH) and ethyl acetate (EtOAc). From the EtOAc, suillin, is reported as the major compound. The cholinesterase inhibitory potential of extracts and the major isolated compound was assessed by Ellman´s method and progression curves were recorded at 405 nm for 60 min. Donepezil hydroclhoride was used as a positive control. The samples were dissolved in methanol at 10 mg/mL and two more 10× dilutions were included to obtain final concentrations of 1, 0.1 and 0.01 mg/mL at the mix of reaction. $IC_{50}$, $K_m$, $V_{max}$, and $K_i$ were calculated for suillin. Suillin (200 mg) along with linoleic acid, ergosterol peroxide and ergosterol were isolated. The EtOH and EtOAc extracts exerted a moderate inhibitory effect ($IC_{50}$ > 200 µg/mL. In adittion, suillin exerted a non-competitive mixed mechanism. against AChE with an $IC_{50}$ value of 31.50 µM and Ki of 17.25 µM. To the best of our knowledge, this is the first report of the anticholinesterase effect of *Suillus luteus* and suillin. The kinetic parameters and the moderate potency of the compound determined in this study, encourage us to propose suillin as a promising chemopreventing agent for the treatment of neurodegenerative diseases such as Alzheimer.

## 1. Introduction

In recent years, research and development of metabolites from plants and macrofungi, with novel structures and biological activities, have gained extensive attention for their potential development as drug candidates [1]. Mushrooms produce a vast diversity of biomolecules with unique chemical structures, interesting nutritional and/or medicinal properties and they have been recognized as functional foods [2–4] In addition, biomedical researches about mushrooms have demonstrated that they are natural sources of bioactive compounds with

**Funding:** The authors recieved no specific funding for this work.

**Competing interests:** The authors have declared that no competing interests exist.

immunomodulatory, antiobesogenic, antioxidant, antiangiogenic, anti-inflammatory, antidiabetic, antimicrobial, and cytotoxic activities [5–8].

*Suillus luteus* (L.) Roussel (Suillaceae) is an edible mushroom commonly called "slippery jack"because their cap cuticle is sometimes slimy: It is also known as "Kallampa" and it is used in traditional medicine to manage gastrointestinal disorders and headaches by Saraguros indigenous people. The Saraguros are one of the Kichwa indigenous communities from southern Ecuador. They have traditionally resided in the northern region of Loja province, with a population of ca. 60000 inhabitants. They are considered one of Ecuador's most organized ethnic communities, preserving their ancestral knowledge, culture, medical and ritual practices, language, and customs. Their main economic activity is focused on agriculture. Their origin is believed according to recent theories to '*mitimaes*' (in kichwa, 'mitmaq', meaning: exiled), a group of indigenous sent by the Inca empire to strategic places to accomplish specific functions as farming, border defense etc., as required. Saraguros have extensive knowledge of plants and their curative properties, based on their health care system where their healers ('Yachak' or 'Hampiyachakkuna') use the curative properties of plants, animals, and/or minerals to threaten several illnesses in the community. Nowadays, Saraguros' culture has experienced essential changes that threaten their customs and preserve their traditional knowledge. [9–11].

Many phytochemical studies from several species of the Suillaceae family such as *S. granulatus* (L.) Roussel, *S. placidus* Bonord, and *S. luteus* reported the tetraprenylphenol suillin as the majority compound [12–14]. Prenylphenols are a large family of compounds including grifolin, neo-grifolin [8], and suillin [15] previously described not only in plants but also in macrofungi. In addition, prenylphenols have been reported to play important physiological roles including cholesterol metabolism regulation, antibacterial, antitumor, antioxidant and anti-inflammatory activities [16–18]. Among the prenylphenols, grifolin neogrifolin and confluentin have been further investigated for their cytotoxic activities. From the ethanol extract of the terrestrial polypore *Albatrellus fletti* (Morse ex Pouzar) Albatrellaceae family, these compounds were isolated and assessed for their cytotoxic potential and found to suppress KRAS expression in human colon cancer cells. Furthermore, confluentin could also induce apoptosis and cell cycle arrest at the G2/M phase in the colon adenocarcinoma SW480 cell line [8].

Tringali et al. [12] isolated suillin from dichloromethane extracts of *S. granulatus* and found that suillin had strong inhibitory effects on human nasopharyngeal carcinoma KB cells, human bronchial cancer nonsmall cell lung cancer (NSCLC)-N6 cells, and particularly, mouse leukemia P-388 cells. Liu et al. [14] also isolated suillin from ethyl acetate extract of *S. placidus* and determined its antitumor spectrum using eight cancer cell lines (HepG2, Hep3B, Huh7, Bcap37, MCF-7, HeLa, H446, and SW620).

Furthermore, from *Suillus luteus*, a new ceramide (suillumide) exerts high potential to inhibit the growth of human melanoma cells SK-MEL-1 with $IC_{50}$ value of 9.7 µM, along with suillin and nine known compounds have been reported [13]. Likewise, from *S. placidus* by countercurrent chromatography, iso-suillin was isolated and demonstrated to be highly cytotoxic against human breast cancer Bcap37. Its mode of action was attributed to the activation of apoptosis [19].

Acetylcholinesterase (AChE) is one of the main cholinesterases in the body. It catalyzes the breakdown of acetylcholine and other choline esters functioning as neurotransmitters. AChE, a cholinergic enzyme, is primarily found at postsynaptic neuromuscular junctions, especially in muscles and nerves, where it can terminate synaptic transmission. AChE inhibition results in accumulation of acetylcholine leading to increased stimulation of muscarinic and nicotinic receptors which provides symptomatic relief to memory deficit in Alzheimer's disease (AD) [20–22]. Consequently, chemical inhibitors that prevents acetylcholine from breaking down by AChE-enzyme, increase both, the level and the duration of action of acetylcholine [23].

Therefore, AChE inhibitors are the most important prescription drugs that medicate early symptoms of AD. To improve the cognitive symptoms of AD, donepezil, galantamine, huperzine A, physostigmine, and tacrine have been developed as AChE inhibitors. However, these medicines have side effects, including vomiting, diarrhoea, body-weight loss, insomnia, and nausea [24, 25]. Hence, it is crucial to search for new AChE inhibitors from natural products without causing side effects.

Interesting natural cholinesterase inhibitors were also found in some mushrooms. Four β-carboline alkaloids, brunneins, and 3-(7-hydroxy-9H-β-carboline-1-yl) propanoic acid were found in the fruiting bodies of the agaricoid fungus *Cortinarius brunneus* (Pers) Fr. (Cortinariaceae) [26]. Brunneins exhibited deficient cholinesterase inhibitory effects and no cytotoxicity [27]. Besides, from *Cyclocybe cylindracea* DC., (Strophariaceae), a new β-carboline brunnein type with free radical scavenging capacity (DPPH, $EC_{50}$ 119.1 μg/mL) has been reported [28]. From, *Cortinarius infractus* Berk (Cortinariaceae), two alkaloids, infractopicrin and 10-hydroxy-infractopicrin, were isolated. Both compounds show AChE-inhibitory activity possessing a higher selectivity than galanthamine [29].

Other natural anti-cholinesterases of the unknown chemical structure have been observed in mushrooms of the Agaricaceae family. The ethyl acetate and n-hexane extract of *Agaricus bitorquis* (Quél.) and the hexane extract of *Agaricus essettei* Bon, showed meaningful anti-BuChE activity, being close to that of galantamine [30]. The methanolic extract of *Pleurotus pulmonarius* (Fr. Quél. (Pleurotaceae), showed moderate inhibitory activity against acetylcholinesterase and butyrylcholinesterase [31]. In a related study, the ethanolic extract of *Pleurotus ostreatus* (Jacq.) P. Kumm. reported high AChE inhibitory activity [32].

The edible mushroom *S. luteus* has not been reported previously as an acetylcholinesterase inhibitory. In this study, we aimed to evaluate the AChE inhibitory activity of the extracts of *S. luteus* and the kinetic profile of the main isolated compound suillin. The results from this study, it might be the supporting data for indicating that *S. luteus* has health benefits as a functional food.

## 2. Materials and methods

### 2.1. General information

Fruit dehydrator Stöckli™ was used to dry mushroom samples before extraction. The organic solvents used for column chromatography (CC) and thin-layer chromatography (TLC) were purchased from Brenntag (Guayaquil, Ecuador) and carefully distilled before using. Silica gel 60 (Merck KGaA, Darmstadt, Germany, from 0.063 to 0.200 mm) was used as a stationary phase for CC. Normal phase TLC (with fluorescence indicator at 254 nm) were purchased from Sigma-Aldrich. After exposure to UV light (254 and 366 nm), the plates were revealed with a mixture of sulphuric acid and vanillin. The sample elucidation process was performed based on the nuclear magnetic resonance (NMR) spectra of [1]H and [13]C at 400 MHz and 100 MHz, using a VARIAN Agilent 400 MHz Premium Shielded with an NMR Y 0021953 magnet and with an MR1005 W031 console and ONE NMR Probe 5 mm. Chemical shifts were reported in δ (ppm), relative to the signal of tetramethylsilane (TMS) and coupling constants (*J*) in Hz. The LR-ESIMS analyses were performed on a Dionex UltiMate3000 modular system coupled with a UV detector with multiple channel recording (Thermo Scientific, Germany) and MS detector Bruker amaZon Ion Trap Mass (Bruker, Germany).

For the cholinesterase inhibitory assays, DTNB (5,5-dithiobis-[2-nitrobenzoic acid]) (99.9%) anticholinesterase compound donepezil-hydrochloride (>99%), the electric eel acetylcholinesterase (AChE, Type-VI-S, EC 3.1.1.7, 137 U/mg protein) and the substrate, acetyltiocholine iodide, (>99%) were purchased from Sigma-Aldrich (St. Luis. MO, USA).

## 2.2. Fungal material

Fruiting bodies of *Suillus luteus* were collected in Loja province, Zamora Huayco sector, southern Ecuador, in May 2015, at 2500 m a.s.l. (Coordinates 4°0′17.1324″S., 79°11′3.39″W). It was correctly identified and authenticated by mycologist Darío Cruz. A voucher specimen (No. JMA054) has been deposited at the Herbarium HUTPL of the Universidad Técnica Particular de Loja. Mushroom samples free of impurities were placed in a fruit dehydrator apparatus at 30°C for 24 h before extraction.

## 2.3. Extraction and isolation

Dry fruiting bodies (500 g) of *Suillus luteus* were extracted three times by maceration at room temperature for 24 hours each time. Ethyl acetate (EtOAc), and ethanol (EtOH) were used in this order at 1:4 w/v ratio (at least 2 liters of each solvent). Following filtration and evaporation of the solvents under vacuum, two dry extracts were obtained: ethyl acetate extract (17.39 g) with a yield of 3.48% w/w and ethanol extract (11.34 g) with a yield of 2.27% w/w. The relative yield was calculated according to the mass of dry extract and the mass of total extracted fungal dry material.

The EtOAc extract (5g) was submitted to column chromatography (CC) on silica gel, using an increasing polarity gradient from 100% n-hexame to 100% EtOAc to afford 17 fractions (Fr1-Fr17). In addition, thin-layer chromatography (TLC) was carried out to monitor the elution performance on CC.

Fraction 9 (303.5 mg) was further submitted to CC on silica gel using an isocratic elution system of n-hex:EtOAc (7:3) yielding 200 mg of a pure compound that by NMR spectral analysis and mass spectra were found to be the tetraprenylphenol suillin (**4**). Mass spectra were recorded in the negative ion mode with an electrospray ion source operating at atmospheric pressure. The electrospray needle was used with a voltage differential of 4–5 kV and a sheath flow of 2 μL/min of a 1:1000 mixture of suillin in MeOH.

## 2.4. Anticholinesterase inhibition assay

The inhibition of AChE was measured using the spectrophotometric method developed by Ellman et al. [33], with slight modifications as suggested by Rhee et al. [34] and fully detailed for our research group in a previous study [35]. Briefly, the reaction mixture contained 40μL of Buffer Tris, 20 μL of the tested sample solution, 20 μL of acetylthiocholine (ATCh, 15 mM, PBS pH 7.4), and 100μL of DTNB (3 Mm, Buffer Tris). Pre-incubation was carried out for 3 min at 25°C and continuous shaking. Finally, the addition of 20 μL of 0.5 U/mL AChE started the reaction, and the amount of product released was monitored in an EPOCH 2 (BIOTEK®) microplate reader at 405 nm, 25°C and 60 min.

Sample solutions of EtOH and EtOAc extracts from *S. luteus* were made by dissolving 10 mg in 1 mL MeOH. Two more dilutions (10 × factor dilution) were included to obtain 1000, 100, and 10 μg/mL final concentrations. All the compounds were tested at a maximum concentration of 250 μM. Progression curves were calculated from absorbance, according to a standard curve of DTNB and L-GSH at different molar concentrations to measure the initial velocity, expressed as mM/min of product released. The corresponding $IC_{50}$ value was calculated by curve fitting data (linear regression or non-linear regression analysis, PRISM 8.0.1, GraphPad, San Diego, CA, USA). As a protic non-selective solvent, MeOH was selected to dissolve samples and employed as a negative control at a maximum concentration of 10% in the final mix volume without affecting enzyme reaction. Donepezil-hydrochloride was used as a positive control with a calculated $IC_{50}$ value of 12.40 ± 1.35 nM close to our previous report, as shown by Valarezo et al. [35].

## 2.5. Suillin kinetic analysis

Kinetic parameters, $K_m$ and $V_{max}$, were determined for suillin by measuring the change in enzyme velocity as a function of the substrate and three concentrations of the inhibitor (I) (calculated $IC_{50}$ served as reference). Progress curves for AChE with 2.5, 5, 10, 15, 20 and 25 mM of ATCh were obtained (final concentrations ranging from 0.25 to 2.5 mM in the mix). Final concentrations of 50, 40 and 30 μM of suillin in MeOH were included in determining the kind of enzyme inhibition. Inhibition constant ($K_i$), the concentration required to produce half-maximum inhibition, was obtained according to the type of inhibition previously determined. The amount of enzyme employed was the same as used in the enzyme inhibition assay above described and followed the same procedure conditions. All the calculations were done with Graph pad prism V8.0.1, non-linear regression, mixed inhibition general kinetic model included in the software pack (1, 2, 3).

$$V_{maxApp} = V_{max}/((1 + [I])/(Alpha \times Ki)) \tag{1}$$

$$K_{mApp} = (Km(1 + [I])/(Ki))/((1 + [I])/(Alpha \times Ki)) \tag{2}$$

$$Y = V_{maxApp}(X)/\left(K_{mApp} + X\right) \tag{3}$$

## 3. Results and discussion

### 3.1. Isolation and characterization

The dried fruiting body of the fungus *S. luteus* was extracted with EtOAc and EtOH but only the EtOAc extract was selected for chromatography because of the feasibility to work on silica-gel with a medium polarity extract. TLC revealed the same compound on both extracts (data not shown) and the anticholinesterase activity for both was similar. Repeated chromatographic separations of the hexane-ethyl acetate soluble fractions led to the purification of suillin **(4)**. It was isolated as a yellow oil (200 mg), and its molecular formula was assigned as $C_{28}H_{40}O_4$ based on LR-ESI-MS (*m/z*,ion at 439.13, [M-H]⁻ calculated for 439.47) spectra and NMR data. The molecule was characterized by spectroscopic techniques (LR-MS and NMR analysis) and further comparison with literature data [36, 37].

Spectral data are described below.

Suillin **(4)**: ¹H-RMN (400 MHz, CDCl₃): δ 6.71 (1H, d, *J* = 8.8 Hz, H-5), 6.50 (1H, d, *J* = 8.8 Hz, H-6), 5.62 (s, 1H), 5.40 (s, 1H), 5.23 (t, *J* = 7.0 Hz, 1H), 5.11–5.06 (m, 3H), 3.26 (d, *J* = 7.0 Hz, 2H), 2.28 (s, 3H), 2.13–2.02 (m, 6H), 1.97 (m, 4H), 1.79 (s, 3H), 1.68 (d, *J* = 0.8 Hz, 3H), 1.60 (d, *J* = 0.8 Hz, 9H).

¹³C-RMN (100 MHz, CDCl₃): 142.97 (C-1), 142.36 (C-2), 112.93 (C-3), 114.03 (C-4), 142.11 (C-5), 120.22 (C-6), 24.15 (C-7), 120.88 (C-8), 139.42 (C-9), 39.77 (C-10), 26.44 (C-11), 123.65 (C-12), 135.19 (C-13), 39.84 (C-14), 26.74 (C-15), 124.30 (C-16), 135.97 (C-17), 39.86 (C-18), 26.90 (C-19), 124.53 (C-20), 131.44 (C-21), 25.84 (C-22), 17.83 (C-23), 16.15 (C-24), 16.22 (C-25), 16.36 (C-26), 170.25 (C-27), 20.96 (C-28).

Linoleic acid **(1)**, ergosterol peroxide **(2)**, and ergosterol **(3)** were additionally isolated from (EtOAc) extract (**Fig 1**). These compounds are known compounds commonly isolated from edible fungi. Linoleic acid **(1)** is a polyunsaturated essential fatty acid found mainly in plant oils. Ergosterol peroxide **(2)** seems to be distributed among fungi natural compounds, too. It has been obtained from some fungi and marine organisms [38, 39]. In some recent studies, peroxyergosterol showed potent antioxidant and anti-inflammatory activities and inhibitory

**Fig 1. Four known isolated compounds isolated from *Suillus luteus*.**

effects on some cancer cell lines [40–44]. Ergosterol (**3**) is the provitamin D2 and vitamin D2 was shown to contribute to the prevention of prostate and colon cancer [45].

### 3.2. Anticholinesterase assays

Half inhibition concentrations required to inhibit the enzyme activity exerted by ethanolic ethyl acetate extract and the major isolated compound, suillin, are summarized in **Table 1**. Both extracts exert the same potency, and it can be attributed to the significant occurrence of the main compounds, which has a moderate inhibitory effect over AChE (**Fig 2**). Linoleic acid (LA), ergosterol and peroxyergosterol did not exhibited inhibitory activity at the maximum dose tested (200μM), however, dietary consumption of LA and other related PUFAs could be associated to a cholinergic transmission improvement, acting as neuroprotective agents in aged brains, where it has been demonstrated that the decrease of fatty unsaturated acids is associated to a decline in the neurological function [46]. To the best of our knowledge, this is the first report of the cholinesterase inhibitory effect exerted by suillin, a potent apoptosis inducer isolated from many fungal species from *Suillus* genus [37].

**Table 1. Half inhibitory concentrations of extracts and the major isolated compound suillin as inhibitors of AChE enzyme.**

| Sample | $IC_{50} \pm SD$ (μg/mL, μM[†], nM[‡]) |
|---|---|
| ethanol extract | 237.95 ± 3.59 |
| ethyl acetat extract | 241.57 ± 5.93 |
| suillin[†] | 31.50 ± 1.03 |
| donepezil hydrochloride | 12.40 ± 1.35 |

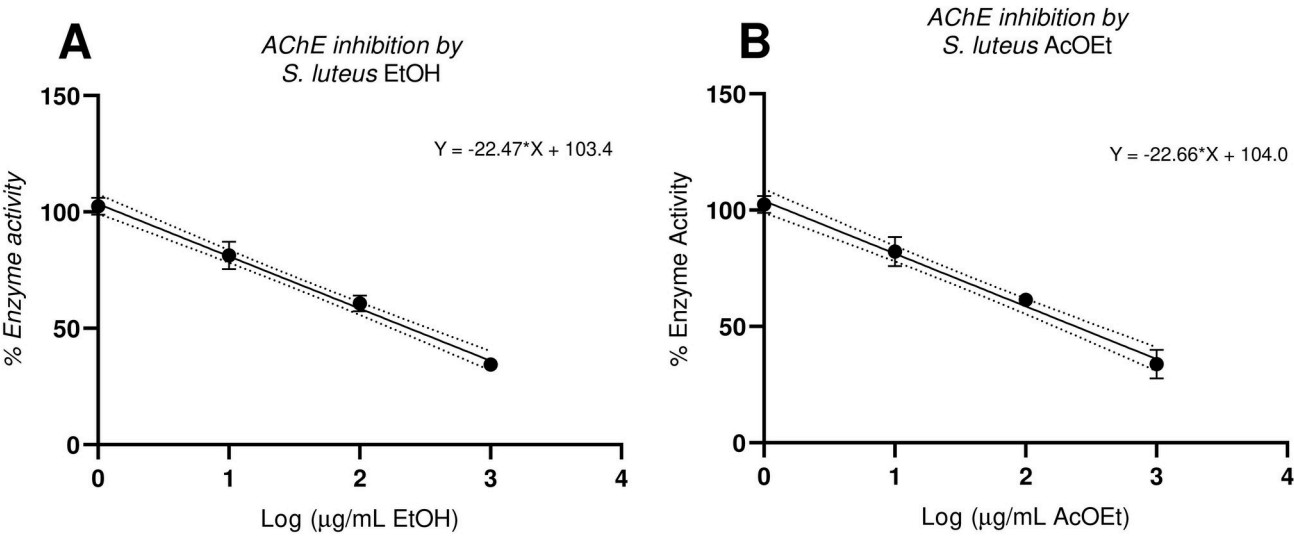

**Fig 2.** AChE inhibition effect exerted by different concentrations of EtOH (A) and AcOET (B) extracts from *Suillus luteus*.

### 3.3. Suillin kinetics

A dose-response curve obtained for three concentrations of suillin can be observed in **Fig 3**. $IC_{50}$ for Suillin was calculated from the nonlinear regression analysis (Normalized response vs. Log [Inhibitor]-variable slope).

In the same way, according to the general mixed model analysis, suillin, exerted a mixed inhibition mechanism where a $K_m$ value appears to decrease and $V_{max}$ is also reduced as a function of the interaction of the inhibitor with the free enzyme (E) or with the complex ES. Some authors usually call non-competitive inhibition (mixed) or mixed competitive when alpha is higher than one but lower than ten, and closely mimics uncompetitive binding but differ from this in the fact that in uncompetitive, the inhibitor exclusively binds to the ES complex with no affinity for free E (alpha is very low but greater than zero) [47]. The theoretical relationship between $K_i$ and $IC_{50}$ for a mixed model can be $K_i = IC_{50}$ to $K_i = \frac{IC_{50}}{2}$ depending on the ratio of $K_{i\alpha}$ (binding affinity of I for free E) to $K_{i\beta}$ (binding affinity of I for ES) [48]. According to our results, the relation between the two parameters mentioned above is close to 1:2 ratio, which could agree with our proposed model. However, our in vitro assay lacks the tools to determine this proposal precisely. All kinetic parameters calculated for suillin are presented in **Table 2**.

Michaelis-Menten kinetic behaviour for three doses of suillin against AChE can be observed in **Fig 4**. Lineweaver-Burk plot was drawn (as inset) only with three points to demonstrate the mechanism of inhibition.

There is no extensive literature for natural cholinesterase inhibitors from mushrooms. To date, only a few compounds have been isolated and tested against acetylcholinesterase. For example, Patočka J. [25], reported four β-carboline alkaloids, bruenins A-C, and 3-(7-hydroxy-9H- β -carboline-1-yl) propanoic acid, isolated from the fruiting bodies of *Cortinarius brunneus*, exhibited a deficient activity. On the other hand, from *Cortinarius infractus*, two alkaloids, infractopicrin and 10-hydroxy-infractopicrin showed inhibitory effects with $IC_{50}$ values of 9.72 and 12.7 μM, respectively. That is in the same range of action exhibited by suillin in the present work, which encourages us to claim that suillin and other

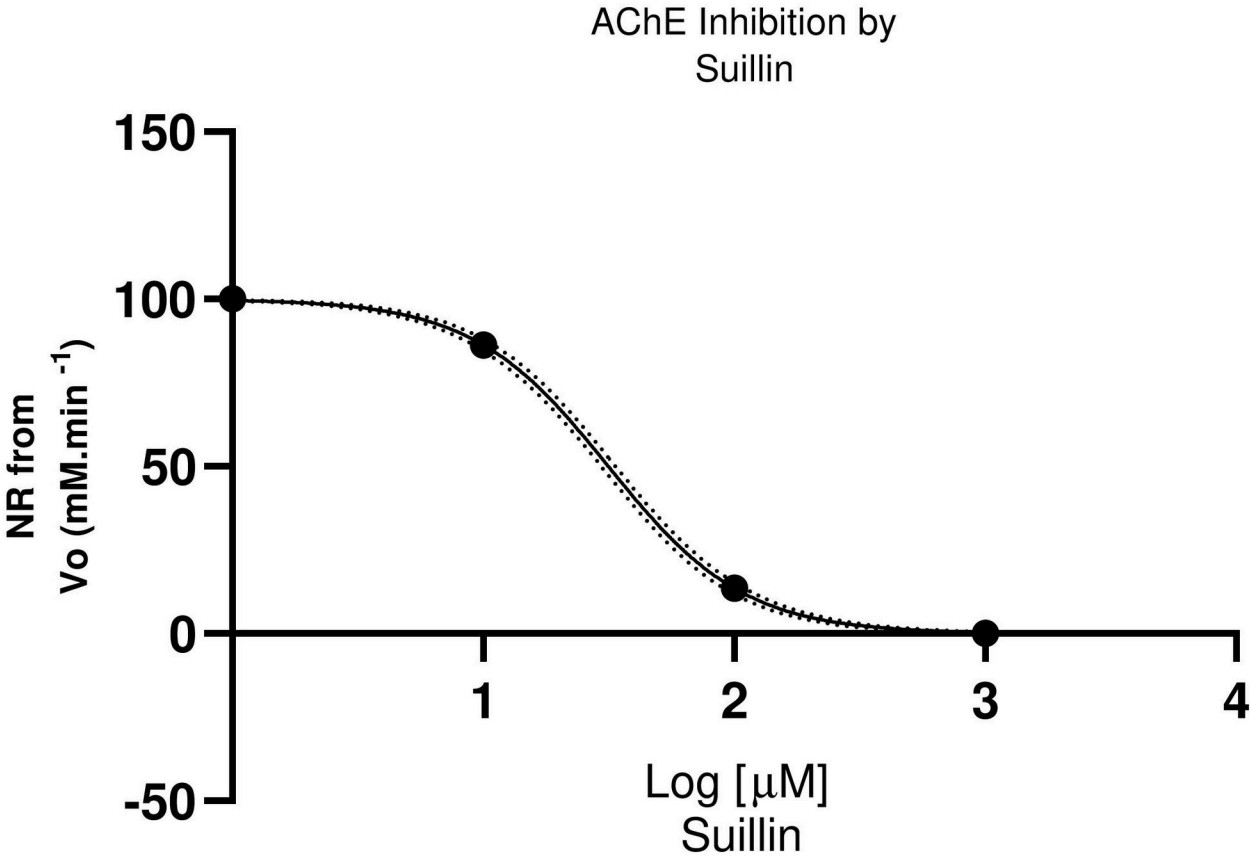

**Fig 3. AChE inhibition effect exerted by suillin, the major isolated compound.** Graph drawn from nonlinear regression model.

isolated compounds from mushrooms can be attractive for the treatment of Alzheimer disease (AD).

In the same context, Deveci et al. [6], reported the isolation of a new steroid 5α,8α-epidioxyergosta-6,22-dien-3β-il-palmitate, along with ten known compounds with a weak profile of inhibition, with inhibition percentages below 15% at 100 μg/mL doses. Thus, it can explain the fact that steroids present a poor inhibition profile as demonstrated for the null effect observed for the remaining isolated compounds against AChE.

**Table 2. Kinetic parameters of suillin determined for AChE.**

| Compound | Dose μM | Vmax ± SD mM/min | kM ± SD μM | Kind of Inhibition | Ki ± SD μM |
|---|---|---|---|---|---|
| control | 0 | 38.34 ± 1.38 | 668.7 ± 7.04 | Mixed inhibition (non pure noncompetitive inhibition) | 17.25 ± 0.67 |
| suillin | 30 | 23.10 ± 1.71 | 1028 ± 18.45 | | |
| | 40 | 14.01 ± 0.72 | 723 ± 12.08 | | |
| | 50 | 7.85 ± 0.69 | 687 ± 26.20 | | |
| Global shared | | 38.70 ± 2.07 | 674.60 ± 10.52 | | |

All data represent the media ± standard deviation of media of three experiments with three replicates, n = 9

α = 1.45. When α > 1, the inhibitor preferentially binds to the free enzyme and the model mimics uncompetitive inhibition.

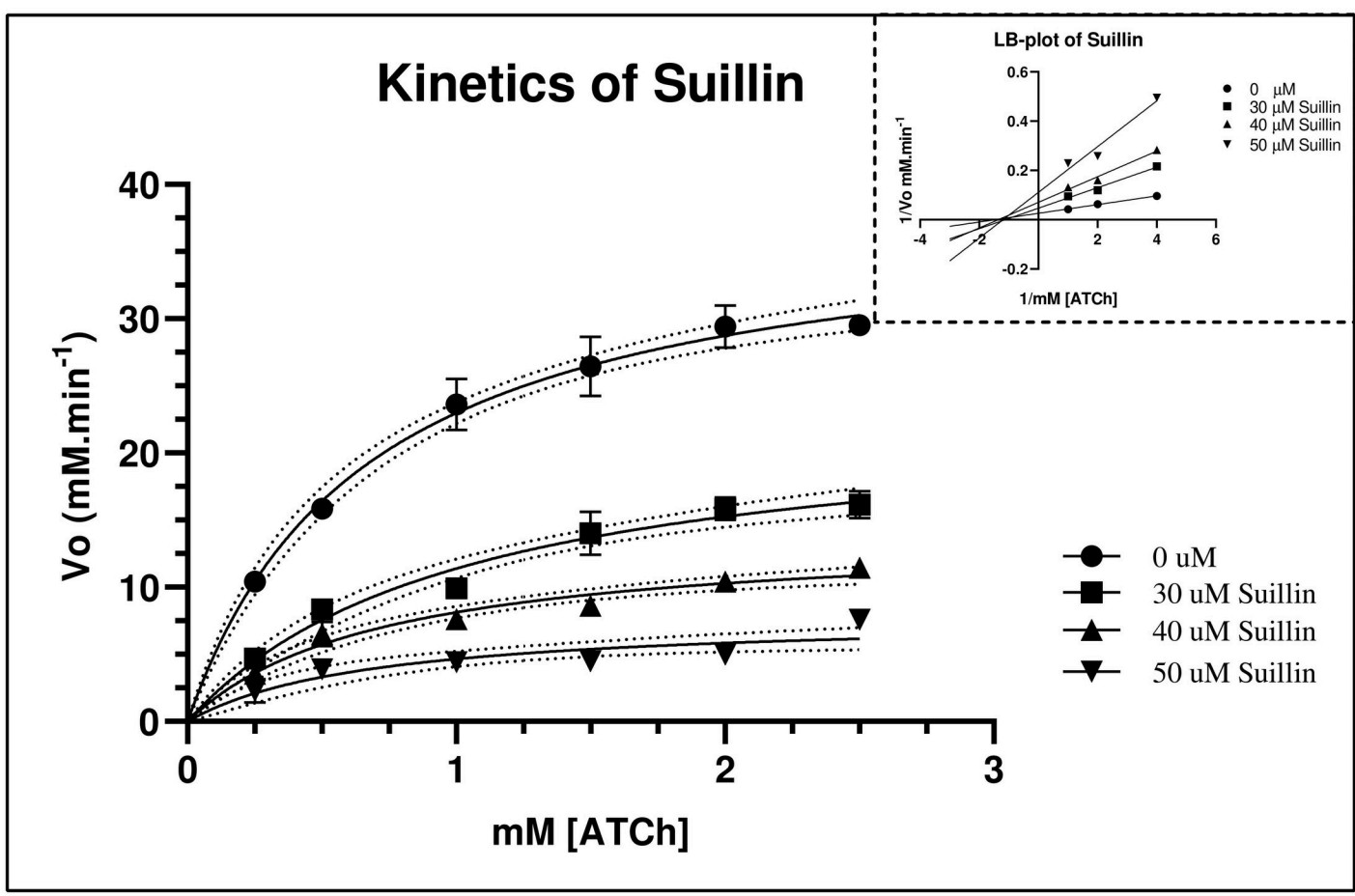

**Fig 4. M-M plot showing variation in Vmax for AChE with three different concentrations of suillin.** Inset. Lineweaver-Burk plot showing noncompetitive effects exerted by suillin.

Suillin exerted a strong cytotoxic effect in 8 human cancer cells, preferentially on human hepatoma HepG2 cells with an $IC_{50}$ of ca. 2μM and mechanistically was found to induce apoptosis in HepG2 cells through both, death receptor and mitochondrial pathway [14]. Suillin has proven to be a promising chemotherapeutic agent for the treatment of liver cancer due to the induction of apoptosis in human cancer cells, through both death receptor pathway and mitochondrial pathway and to exert strong cytotoxic effect against human hepatoma HepG2 cells [9]. The potential of suillin to threaten liver cancer and the anticholinesterase activity determined in this study suggest that this compound could be a promising agent for the pharmaceutical industry.

In Ecuador, a few studies have been conducted about edible mushrooms and traditional practices by communities. For example, in a recent work, performed by Gamboa-Trujillo et al. [49], there is relevant information about the use of 37 edible species of macrofungi, their preparation for consumption and myths. Still, nothing is said about the chemical and pharmacological profile of mushrooms which encourages us to pursue this research field and validate the ethnomycological information given by many communities along the entire country.

Suillin could be an attractive compound for the development of a chemopreventive agent for AD due to its moderate inhibitory capacity, mode of action, high yield in fungal material, the feasibility of collection of the mushroom because it grows extensively associated to *Pinus*

trees but, further studies should be conducted to precise the mechanism of inhibition exerted against AChE. *In silico* studies as docking and in vivo measurements of the effect of suillin on AChE should be assessed to ensure its efficacy and validate the same mode of action. *In vitro* anticholinesterase measurement as performed in this study lacks the reliability of an in vivo model. Still, it gives critical information to continue researching a chemical candidate for a pharmaceutical or nutraceutical preparation that local communities use.

Like an acylesther, suillin could be acting as a substrate for AChE, however, the spectrophotometric method used in this research is useless to measure this kind of activity, in contrast to the method used by Rush et al [50] who determined the effect of aprophen over AchE and Butyrylcholinesterase (BuChE) and found that it acts as a potent reversible inhibitor and a poor susbtrate for BuChE. The method uses a radiolabel [$^{14}$C]. Apropehn synthetic compound and, the enzyme activity was determined from the radioactive decay of radiolabeled product released.

Another possible explanation to the observed effect for suillin could be related to the peripheric anionic site (PAS) of acetylcholinesterase, a second substrate binding site, composed of five residues (Tyr70, Asp72, Tyr121, Trp279, and Tyr334) that lies at the entrance to the active site gorge and enhance the catalysis of Acetylcholine by trapping it to its way to the active site [51]. Suillin could be interacting with this site generating an ESI complex. Several compounds have been demonstrated to interact via PAS site, such as, edrophonium, which exerts a competitive reversible inhibition at a concentration of 0.1 μM and a mixed competitive-nonconmpetitive inhibitor at higher concentrations (1 to 10 μM) [52].

According to literature, 155 families of fungi are represented in Ecuador, particularly species from the order Agaricales, Polyporales, Boletales, and Russulales. Many of them are edible and are considered excellent sources of nutraceuticals that can be assessed.

## Supporting information

**S1 File. Rates of enzyme reaction and enzymatic model.**
(PDF)

## Acknowledgments

The authors thank to Universidad Técnica Particular de Loja and its Chemistry Department for providing the facilities to carry out this research.

## Author Contributions

**Conceptualization:** José Miguel Andrade, Luis Cartuche.

**Formal analysis:** José Miguel Andrade, Luis Cartuche.

**Investigation:** José Miguel Andrade, Pamela Pachar, Luisa Trujillo, Luis Cartuche.

**Methodology:** Pamela Pachar, Luisa Trujillo, Luis Cartuche.

**Project administration:** Luis Cartuche.

**Writing – original draft:** José Miguel Andrade, Luis Cartuche.

**Writing – review & editing:** José Miguel Andrade, Luis Cartuche.

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
