## [Decision Letter · Decision Letter 0]

30 Sep 2021

PONE-D-21-28384Suillin: a strong cholinesterase inhibitor from Suillus luteus, an edible basidiomycete used by Saraguro indigenous, south of EcuadorPLOS ONE

Dear Dr. Cartuche,

Thank you for submitting your manuscript to PLOS ONE. After careful consideration, we feel that it has merit but does not fully meet PLOS ONE’s publication criteria as it currently stands. Therefore, we invite you to submit a revised version of the manuscript that addresses the points raised during the review process.

In your revised manuscript please address as fully as possible the criticisms of both reviewers, especially the severe concerns raised by Reviewer 2.

We look forward to receiving your revised manuscript.

Kind regards,

Israel Silman

Academic Editor

PLOS ONE

Journal Requirements:

"The authors want to thank for the financial support given by Universidad Técnica Particular de Loja. "

"The authors recieved no specific funding for this work"

Reviewers' comments:

Reviewer's Responses to Questions

**Comments to the Author**

1. Is the manuscript technically sound, and do the data support the conclusions?

Reviewer #1: Partly

Reviewer #2: Partly

2. Has the statistical analysis been performed appropriately and rigorously? 

Reviewer #1: I Don't Know

Reviewer #2: Yes

3. Have the authors made all data underlying the findings in their manuscript fully available?

Reviewer #1: Yes

Reviewer #2: Yes

4. Is the manuscript presented in an intelligible fashion and written in standard English?

Reviewer #1: Yes

Reviewer #2: Yes

5. Review Comments to the Author

Reviewer #1: This manuscript describes the isolation, structural analysis and action on acetylcholinesterase of suillin, a terpenoid (tetraprenylphenol) from the basidiomycete Suillus luteus. Suillin is a reversible non-competitive inhibitor of acetylcholinesterase. This compound is interesting from a mechanistic point of view, but likely not for a possible interest as anti-AD drug. However, the study is weak and the enzymologic part needs to be carefully reworked.

Major points

Title has to be changed: suillin is not a strong cholinesterase inhibitor (cf line 231, Table I, Ki = 22 microM)

Introduction: Line 70, AChE is not only mainly found at neuromuscular junction but also in brain where it plays a key role.

Line 71-72 how high level of AChE at neuromuscular junctions can cause neurological disorders ?

Formula of suillin, Line 140: suillin is an acetylester, like aspirin and acetylcholine. Why the authors did not check whether this compound could be also substrate of acetylcholinesterase ? There are many examples of compounds acting both as a substrate and inhibitor of cholinesterases (ex, aprophen).

Lines 163 and 235 and Fig 4: AChE kinetics at concentrations varying between 2.5 and 25 mM cannot be described by the Michaelis-Menten model. Fig 4 shows the rate of AChE as a function of ATC ranging from 0.25 to 2.5 mM. The enzyme should be inhibited by excess substrate beyond 1 mM. How to explain that the enzyme is not inhibited by excess substrate ?

Line 170: this model equation suppose that the inhibitor only affects Vmax. How to rule out that inhibitor does not affect Km or both Km and Vmax?

Lines 221, 23: IC50 and Ki do not match as they should for pure non competitive inhibitor (as it is shown in Fig 4 insert, alpha =0): Ki = IC50.

Minor points

Line 146 : Was MeOH the solvent of suillin for inhibition measurements ? (What was the final MeOH concentration in assay ?)

Typos

Line 21 : acetylcholinesterase

Line 146 : mM

Reviewer #2: In this manuscript, the authors described their findings on suillin, the active ingredient extracted from Suillus luteus, can function as a cholinesterase inhibitor. The methods adopted in this study included thin-layer chromatography (TLC) to determine compounds from extracts of Suillus luteus, Ellman assay to characterize the extracts’ and suillin’s roles in inhibiting AChE and to conduct the kinetic analysis of suillin. As a conclusion, they claimed that suillin can be a candidate for treating neurodegenerative diseases.

Major comments

1. Table 1 & Figure 2: Positive control for Ellman assay is missing. A known drug in inhibiting AChE activity should be included.

2. As several compounds were identified in the extracts. Why suillin was the major component in suppressing AChE? For other compounds identified, same experiments (i.e. Ellman assay and kinetic analysis) should also be conducted in order to compare and confirm suillin as the main functioning chemical.

3. Many fungi have been found to have anti-cholinesterase function in previous studies, so the perspective of this study is not new. Also, if the authors would like to propose it as a candidate for neurodegenerative treatment, the scale of study in this manuscript is not adequate. More studies (e.g. in vitro) should be conducted to make the findings more reliable and persuasive.

Minor comments

1. Some typos need to be rectified. (e.g. ‘acethylcholinesterase’)

6. PLOS authors have the option to publish the peer review history of their article (what does this mean?). If published, this will include your full peer review and any attached files.

Reviewer #1: No

Reviewer #2: No

---

## [Author Response · Author response to Decision Letter 0]

25 Mar 2022

Dear Editor and reviewers

All the suggestions as requested by the reviewers and editor has been adressed in the 'Revised Manuscript with Track Changes' file that has been subbmited at the Plos One system, for your consideration. All the responses to reviewers were included in a document attached in the submission system of Plos One

Regards!

---

## [Decision Letter · Decision Letter 1]

13 Apr 2022

PONE-D-21-28384R1Suillin: a mixed-type acetylcholinesterase inhibitor from <suillus luteus=""> which used by Saraguros indigenous, southern Ecuador</suillus>PLOS ONE

Dear Dr. Cartuche,

Thank you for submitting your manuscript to PLOS ONE. After careful consideration, we feel that it has merit but does not fully meet PLOS ONE’s publication criteria as it currently stands. Therefore, we invite you to submit a revised version of the manuscript that addresses the points raised during the review process.

In your revised manuscript please address the very minor concerns raised by the two reviewers.

We look forward to receiving your revised manuscript.

Kind regards,

Israel Silman

Academic Editor

PLOS ONE

Journal Requirements:

Reviewers' comments:

Reviewer's Responses to Questions

**Comments to the Author**

1. If the authors have adequately addressed your comments raised in a previous round of review and you feel that this manuscript is now acceptable for publication, you may indicate that here to bypass the “Comments to the Author” section, enter your conflict of interest statement in the “Confidential to Editor” section, and submit your "Accept" recommendation.

Reviewer #1: All comments have been addressed

Reviewer #2: (No Response)

2. Is the manuscript technically sound, and do the data support the conclusions?

Reviewer #1: Partly

Reviewer #2: Yes

3. Has the statistical analysis been performed appropriately and rigorously? 

Reviewer #1: Yes

Reviewer #2: Yes

4. Have the authors made all data underlying the findings in their manuscript fully available?

Reviewer #1: Yes

Reviewer #2: Yes

5. Is the manuscript presented in an intelligible fashion and written in standard English?

Reviewer #1: Yes

Reviewer #2: No

6. Review Comments to the Author

Reviewer #1: The manuscript was greatly improved. However, line 216 page 10, the formula used for determination of Ki is wrong. Is it a typo or a real mistake? Anyway, with such a "mistake" there is still doubt about the values of alpha and Ki.

Reviewer #2: In this manuscript, the authors described their findings on suillin, the active ingredient extracted from the herb Suillus luteus can function as a mixed-type acetylcholinesterase inhibitor. The methods adopted in this study included thin-layer chromatography (TLC) to determine compounds from extracts of Suillus luteus, Ellman assay to characterize the extracts’ and suillin’s roles in inhibiting AChE and to conduct the kinetic analysis of suillin. As a conclusion, they proposed that suillin can be a candidate for treating neurodegenerative diseases. Although the most of problems have been addressed, there are still some questions:

Major comments

- Table 1: IC50 of other chemicals isolated (i.e. linoleic acid, ergosterol etc.) should be included for comparison. What is the extraction efficiency of suillin from the herbal extract?

- In the discussion, how mixed inhibitors to AChE can be discussed. Propose the mechanism of how suillin works on AChE, e.g. docking analysis.

7. PLOS authors have the option to publish the peer review history of their article (what does this mean?). If published, this will include your full peer review and any attached files.

Reviewer #1: No

Reviewer #2: No

---

## [Author Response · Author response to Decision Letter 1]

24 Apr 2022

April, 2022

Emily Chenette

Editor in Chief

Public Library of Science PLOS One 

Subject: Online Manuscript Submission 

To Editor

 I am submitting a required revision for consideration in PlosOne with the changes made in the manuscript, highlighted with the track changes mode, for all the suggestions made by reviewers as described below:

Reviewer 1:

The manuscript was greatly improved. However, line 216 page 10, the formula used for determination of Ki is wrong. Is it a typo or a real mistake? Anyway, with such a "mistake" there is still doubt about the values of alpha and Ki.

Answer:

Formulas used in Graphpad packages for mixed model inhibition were included in the manuscript. There was a mistake not reviewing the formula when we submitted the last version of the document.

Reviewer #2: In this manuscript, the authors described their findings on suillin, the active ingredient extracted from the herb Suillus luteus can function as a mixed-type acetylcholinesterase inhibitor. The methods adopted in this study included thin-layer chromatography (TLC) to determine compounds from extracts of Suillus luteus, Ellman assay to characterize the extracts’ and suillin’s roles in inhibiting AChE and to conduct the kinetic analysis of suillin. As a conclusion, they proposed that suillin can be a candidate for treating neurodegenerative diseases. Although the most of problems have been addressed, there are still some questions:

• Table 1: IC50 of other chemicals isolated (i.e. linoleic acid, ergosterol etc.) should be included for comparison. What is the extraction efficiency of suillin from the herbal extract?

Answer:

Minor components were assessed at a maximum final concentration of 200 µM and no inhibitory effect was observed. To obtain IC50 values of such compounds we would need sample solutions with concentrations higher than 2000 µM and for pharmaceutical purposes there is no relevance to assess them. However, we included some additional information in the manuscript for Linoleic acid as a neuroprotective agent to not dismiss the presence of it as an important isolated compound.

Line 255 states … Linoleic acid (LA), ergosterol and peroxyergosterol did not exhibited inhibitory activity at the maximum dose tested (200µM), however, dietary consumption of LA and other related PUFAs could be associated to a cholinergic transmission improvement, acting as neuroprotective agents in aged brains, where it has been demonstrated that the decrease of fatty unsaturated acids is associated to a decline in the neurological function [46].

• In the discussion, how mixed inhibitors to AChE can be discussed. Propose the mechanism of how suillin works on AChE, e.g. docking analysis.

A hypothesis of how suillin could be acting was included at the end of the manuscript with proper references. Docking analysis (Line 339) had been previously suggested for us to validate the mechanism of action proposed in our study but is matter of another possible future research. In vivo studies are also suggested.

Line 352 states. 

Another possible explanation to the observed effect for suillin could be related to the peripheric anionic site (PAS) of acetylcholinesterase, a second substrate binding site, composed of five residues (Tyr70, Asp72, Tyr121, Trp279, and Tyr334) that lies at the entrance to the active site gorge and enhance the catalysis of Acetylcholine by trapping it to its way to the active site [51]. Suillin could be interacting with this site generating an ESI complex. Several compounds have been demonstrated to interact via PAS site, such as, edrophonium, which exerts a competitive reversible inhibition at a concentration of 0.1 µM and a mixed competitive-nonconmpetitive inhibitor at higher concentrations (1 to 10 µM) [52].

The authors support this reviewed version of the manuscript, and would like it to be appreciated in this important scientific journal. It is our first interaction with this journal and we would be very grateful if A. M. Abd El-Aty (orcid.org/0000-0001-6596-7907, Cairo University, Egipt) participate as Academic editor four our work.

Yours sincerely, 

Prof Dr Luis Emilio Cartuche Flores 

Departamento de Química 

Universidad Técnica Particular de Loja, 

P.O. Box 11 01 608, Loja - Ecuador, 

Tel: +593 7 370 1444, 

E-mail address: lecartuche@utpl.edu.ec

---

## [Editor Report · Decision Letter 2]

27 Apr 2022

Suillin: a mixed-type acetylcholinesterase inhibitor from <suillus luteus=""> which used by Saraguros indigenous, southern Ecuador

PONE-D-21-28384R2</suillus>

Dear Dr. Cartuche,

We’re pleased to inform you that your manuscript has been judged scientifically suitable for publication and will be formally accepted for publication once it meets all outstanding technical requirements.

Kind regards,

Israel Silman

Academic Editor

PLOS ONE
---

## [Editor Report · Acceptance letter]

6 May 2022

PONE-D-21-28384R2 

Suillin: a mixed-type acetylcholinesterase inhibitor from *Suillus luteus* which is used by Saraguros indigenous, southern Ecuador 

Dear Dr. Cartuche:

I'm pleased to inform you that your manuscript has been deemed suitable for publication in PLOS ONE. Congratulations! Your manuscript is now with our production department. 

Kind regards, 

on behalf of

Prof. Israel Silman 

Academic Editor

PLOS ONE